# Sociodemographic Predictors Associated with the Willingness to Get Vaccinated against COVID-19 in Peru: A Cross-Sectional Survey

**DOI:** 10.3390/vaccines10010048

**Published:** 2021-12-30

**Authors:** David Vizcardo, Linder Figueroa Salvador, Arian Nole-Vara, Karen Pizarro Dávila, Aldo Alvarez-Risco, Jaime A. Yáñez, Christian R. Mejia

**Affiliations:** 1Facultad de Medicina, Universidad Peruana de Ciencias Aplicadas, Lima 15023, Peru; david.arturoalba@hotmail.com (D.V.); linderfyb@yahoo.com (L.F.S.); arian_nole@hotmail.com (A.N.-V.); karenpizarrodavila@gmail.com (K.P.D.); 2Facultad de Ciencias Empresariales y Económicas, Carrera de Negocios Internacionales, Universidad de Lima, Lima 15023, Peru; aralvare@ulima.edu.pe; 3Vicerrectorado de Investigación, Universidad Norbert Wiener, Lima 15072, Peru; 4Gerencia Corporativa de Asuntos Científicos y Regulatorios, Teoma Global, Lima 15073, Peru; 5Translational Medicine Research Centre, Universidad Norbert Wiener, Lima 15073, Peru; christian.mejia.md@gmail.com

**Keywords:** COVID-19, Peru, vaccine, vaccination, coronavirus, SARS-CoV-2, vaccines, pandemic

## Abstract

During the race for the development of a vaccine against COVID-19, even before its commercialization, part of the population has already shown a growing fear of its application. We designed an analytical cross-sectional study using an anonymous survey in the 25 departments of Peru. We surveyed whether the participants were planning on getting vaccinated, as well as other characteristics that were cross-checked in a uni-, bi- and multivariate manner. Of the 1776 respondents, 70% (1251) stated that they were planning to be vaccinated, 20% (346) did not know yet or doubted it, and 10% (179) did not want to be vaccinated. We observed that those who did not get infected with COVID-19 exhibited a higher frequency to not wanting or were uncertain about getting vaccinated (aPR: 1.40; 95% CI: 1.09–1.81; *p*-value = 0.008). In contrast, there was a lower frequency of vaccine refusal among university students (aPR: 0.75; 95% CI: 0.61–0.92; *p*-value = 0.005) and healthcare workers (aPR: 0.59; 95% CI: 0.44–0.80; *p*-value = 0.001); adjusted by place of residence. There is still an important percentage of respondents who do not want to be vaccinated or are hesitant to do it, which was associated with educational level, being a healthcare worker and if they were previously infected with COVID-19. Our results could offer useful information about COVID-19 vaccination campaigns.

## 1. Introduction

Since December 2019, the world changed by the COVID-19 pandemic with over 258 million confirmed cases, over 5.1 million deaths and over 7.4 billion vaccine doses administered worldwide as of 23 November 2021 [1]. This disease is characterized by progressive and severe pneumonia, with characteristic symptoms such as fever, dyspnea, dry cough, fatigue, headache, anosmia and ageusia [2,3,4]. The first confirmed case in Peru was reported on 8 March 2020 [5], and the number of cases rapidly increased despite the measures established by the Peruvian government [6,7]. Multiple publications have illustrated the fragmented healthcare system in Peru [8,9,10,11,12,13,14,15,16], which has not been the most effective during the COVID-19 pandemic, resulting in a high number of physicians’ deaths [17], limited public policies [18] and detrimental effects to the mental status of the population [19,20,21] and the increase in technostress in university students [22]. Furthermore, Peru has reported discrepancies in the official reports of COVID-19 deaths nationwide [23], poor execution of SARS-CoV-2 testing and reporting [24] and an increased number of COVID-19 cases in children and adolescents [6,25].

Physical isolation was the main preventive measure implemented worldwide to avoid contagion [6,26,27], which caused multiple lifestyle changes in people. Many people have experienced fear of being infected while experiencing the death of family members and friends [28], which has resulted in anxiety and mental distress [29,30]. The widespread disinformation [5,31], fake news [7] and anti-vaccine comments [32,33] have caused fear in the population who seek solutions to prevent or alleviate the symptoms of the disease, since they feel the only resource available is to self-help, self-care and self-medicate [34]. Therefore, it has been reported that some people resort to self-medication [35,36] and others to the use of medicinal plants [37,38] as potential but unproven methods to ameliorate and/or prevent symptoms related to COVID-19. Many have urged that the general state of disinformation be addressed by governmental institutions [39,40,41].

The Peruvian government implemented various regulations to control the purchases of vaccines from different pharmaceutical companies. For instance, the Ministry of Health in Peru made the first publication regarding a vaccine against COVID-19 in September 2020 in the resolution No. 686-2020 to establish guidelines regulations on vaccine research as a result of the COVID-19 outbreak [42]. Then, the first legal standard related to the COVID-19 vaccine was published in October 2020, through ministerial resolution No. 848-2020 with the objective of implementing safe vaccination as a preventive measure against COVID-19 in the country, through the provision of safe and quality vaccines, their administration and proper management [43]. Additionally, this resolution introduced the different Vaccination Phases I, II and III, where the target populations and other measures were indicated to be able to implement the plan [43]. Phase I had the objective of protecting the integrity of the healthcare system by vaccinating the healthcare personnel working in the public and private sector, personnel from the armed forces and police, firefighters, Red Cross, security personnel, brigade members and cleaning personnel and students of health-related careers first [43]. This was followed by Phase II, with the objective of reducing severe morbidity and mortality in the population at greatest risk, which included adults over 60 years, people with comorbidities, the population of native or indigenous communities and personnel working at the National Penitentiary Institute and incarcerated people [43]. Finally, Phase III had the objective of reducing the transmission of infection in continuity and generating herd immunity, which included people between 18 and 59 years of age [43]. An official schedule detailing the projection to vaccinate 27.4 million Peruvians with the two doses was announced, including children between ages 12 to 18 [44]. Furthermore, starting on 26 November 2021 a third vaccine dose can be administered to adults over 18 years who received the second dose at least 5 months before [45].

The National Center for Supply of Strategic Health Resources (CENARES) was assigned to execute the purchase agreement with the company Sinopharm to purchase the first approved vaccine in Peru on January 2021 [46], followed by AstraZeneca on the same month [47], Pfizer on February 2021 [48] and additional Pfizer and AstraZeneca vaccine doses through COVAX facility on March 2021 [49]. Peru has approved, through its regulatory agency DIGEMID, four out of the eight vaccines approved for emergency use around the world [50]. The first vaccine against COVID-19 to receive the conditional sanitary registration by DIGEMID in Peru was Pfizer, which occurred on February 2021, allowing for its import and use in vaccination campaigns [51]. On April 2021, the AstraZeneca vaccine obtained permission from DIGEMID for the active immunization of Peruvians over 18 years of age [52]. On July 2021, Johnson & Johnson’s Jannsen vaccine was approved by DIGEMID [53]. The last of the vaccines to obtain conditional health registration in Peru was Sinopharm’s COVID-19 vaccine on August 2021 [54]. Regardless of its late approval, this was the first vaccine to be applied in Peru after it received exceptional use to import of one million doses [54].

With all the purchases going as planned to cover the vaccination of population of Peru, some changes were made to the National Vaccination Plan against COVID-19 in Peru, adding that Phase I must include healthcare workers [55]. Then a high-level consultative team was formed to recommend the criteria and ethical considerations in decision-making regarding the prioritization of groups to be vaccinated during the execution of the National Vaccination Plan against COVID-19 [56]. With the onset of the COVID-19 pandemic in Peru, it was decided to regulate the framework of clinical trials by implementing rapid ethical review and supervision processes through the creation of the Transitional National Research Ethics Committee for evaluation and supervision of COVID-19 Clinical Trials (CNTEI-COVID-19) [57]. However, these ethical measures were violated with the political scandal called Vacunagate that involved 470 people inoculated outside the clinical trial with the Sinopharm vaccine before it was approved and was available for emergency use in Peru [58,59]. The list of those secretly vaccinated included the President of the Peru and his wife, senior officials from the Ministry of Health and the Ministry of Foreign Affairs, authorities from the two universities where the trial was conducted (Universidad Peruana Cayetano Heredia and Universidad Nacional Mayor de San Marcos), the team of study researchers, teachers and researchers from both universities and even a representative of the Catholic Church [58,59]. This event gave rise to public protests and anger among the Peruvian population because the clinical malpractice and the evident preference for the political elite regardless of the critical situation the country as suffering with a high number of deaths and the economic impact in Peru, exposing again the corruption in the country [58,59,60,61,62]. The alarming and unethical actions of the inoculated professionals caused distrust of the clinical trial that was carried out, the educational institutions involved and the need for vaccination by the population [62]. As an immediate action, the National Institute of Health ensured future protection by forming a National Bioethics Commission to create norms and establish sanctions to avoid further breaches of medical ethics and improve training in ethics and scientific integrity for researchers [63].

Since the beginning of the pandemic, possible vaccines have been developed by different laboratories. Thus, in December 2020, the first vaccine, Pfizer/BioNTech, was approved in the United Kingdom, initiating the vaccination of front-line healthcare personnel and at-risk populations such as the elderly, a process that was followed by different countries [64]. However, during the race for the development of a vaccine against COVID-19, even before its commercialization, part of the population already showed a growing fear of receiving it. This fear was justified by the lack of knowledge of the adverse effects and complications, the lack of confidence in the process, due to the reduced research time and some myths such as the insertion of chips, among others [65,66,67,68]. Despite the fact that Peru was in the first place with the most deaths from COVID-19 during 2020, a rejection attitude towards vaccines against the disease was observed. This reduced acceptance of vaccination was due to the high prevalence of fear of adverse effects that the country was facing, reaching 90.5% [69]. A survey carried out in January 2021 showed us that since August of the previous year, there was an increase in people who did not plan to be vaccinated against COVID-19, going from 22 to 48%, justified by distrust of the countries that manufacture the vaccines, fear of adverse effects and even a preference for being treated with ivermectin [70]. However, thanks to political corruption scandals due to clandestine vaccination in the Vacunagate, a loss of this prevalence of fear of adverse effects has been achieved [58,69]. The importance of this issue is reflected in the vaccination process itself, especially after knowing that some sectors of the population refuse to get vaccinated in countries where the process has already begun [71]. In Peru, vaccination is already underway for people over 12 years old and a third vaccine dose is been administered to adults over 18 years who received the second dose at least 5 months before. As of 26 November 2021, 17.8 million people in Peru have received both doses of the COVID-19 vaccine, constituting about 65% of the intended 27.4 million people to be vaccinated [50]. Therefore, it is necessary to determine the factors associated for people in Peru not wanting to be vaccinated. The general objective of our study was to determine the sociodemographic predictors associated with the willingness of getting vaccinated against COVID-19 in Peru.

## 2. Materials and Methods

### 2.1. Design and Population

An analytical cross-sectional study was carried out in the 25 departments of Peru. A self-administered virtual survey was conducted as shown in Appendix A. The survey was initially evaluated by 10 expert judges using Aiken’s V [72]. After including the experts’ observations, a pilot study was performed in the second week of November 2020 in the 25 departments of Peru. The pilot data were used to calculate the minimal sample size necessary for the actual study. We utilized non-probability snowball sampling [73]. The necessary sample size was calculated using the determinants related to the acceptance of the COVID-19 vaccine in the United States and Southeast Asia by extracting the percentage of vaccine acceptance and the confidence interval [74,75]. For Peru, it was determined that the minimum sample size was 385 individuals, with a prevalence of vaccine acceptance of 50%, a 95% confidence interval and a margin of error of 5%. This was calculated with Epidat software (V.4.2.). To this end, personal data, education, work situation, comorbidities, exposure to the SARS-CoV-2 virus, knowledge about COVID-19, perception of risk about the new COVID-19, sources of information and preventive measures were taken into account.

### 2.2. Variables

The primary variable was whether they would agree to receive the COVID-19 vaccine. We utilized a previously published instrument [75], which was adapted and translated to Spanish. In addition, the instrument was culturally validated through a report that evaluated each question from the survey on three criteria: relevance, coherence and clarity. The report was filled by a specialist in Social Sciences and then the survey was sent to the population. The survey consisted of 2 groups of questions: demographic questions and knowledge about COVID-19.

For the covariables, we considered the sociodemographic variables (age, gender, education, marital status, department of residence, having a chronic disease, household size, work situation) and knowledge about coronavirus (being sick with COVID-19, the reason for getting vaccinated against COVID-19, knowledge about the COVID-19 pandemic and level of knowledge about COVID-19).

### 2.3. Procedures

The actual survey used for the study was adapted virtually, in order to reach as many participants as possible in the different departments, due to the Peruvian state of health emergency. The survey was shared between the second week of December 2020 and the third week of January 2021 through social networks (WhatsApp, Facebook and Instagram) with the objective of reaching the population over 18 years old of the 25 departments of Peru. According to the National Institute of Statistics and Informatics (INEI), the last census conducted in 2017 showed that five departments comprised more than half of the country’s population [76]. These were Lima, Piura, La Libertad, Arequipa and Cajamarca; it was prioritized to take more surveys in these regions. In order to do this, the researchers contacted healthcare personnel to ask for collaboration and dissemination of the survey, providing informed consent where the confidentiality of the information is assured. The survey was available to the participants who accepted the informed consent, voluntarily participated in the study and authorized the use of their data for the present study.

### 2.4. Data Analysis

Each survey was entered into a database in Microsoft Excel 2013, where quality control was performed by reviewing the complete and correct completion of the surveys. This was carried out with Stata v.14 statistical software. Then, a descriptive analysis of the categorical variables was made, obtaining frequencies and percentages; for the quantitative variable, the normality of the data was analyzed and described with central tendency and dispersion measures. A figure was also prepared to show the percentages of each of the categories of the three dependent variables.

Subsequently, a bivariate analysis was generated, where the dependent variables was crossed with each independent variable, for which the *p*-values were obtained in each case. Then, for the multivariate statistics, the variables that were statistically significant in the bivariate model were crossed in each case. For this analysis the generalized linear models were used (with the Poisson family, the log function, models for robust variances and adjusted by each respondent site). The adjusted prevalence ratios, the 95% confidence intervals (95% CI) and the *p*-values were obtained. In each one of the crossings, the *p*-values < 0.05 were considered statistically significant.

### 2.5. Ethical Aspects

The project followed the guidelines for ethical aspects. The research protocol was approved by the ethics committee of the Universidad Peruana de Ciencias Aplicadas (approval code PI 393-20). The participants were selected to answer the scientific questions, respecting their ideology, identity, judgment, prejudices and other relevant events of the interviewee. The individuals participated in the research in an informed, anonymous and voluntary manner. In addition, in order to conduct the survey, an informed consent was requested. Finally, in order for this study to be solely for research purposes, no participant data were disclosed.

## 3. Results

A total of 1834 people, residents of the 25 departments of Peru, were invited to participate in the survey, of which 1776 were eligible for the survey. Fifty-eight participants were not taken into account because they were repeated or the provided data lacked coherence. Those eligible for the survey were 18 years or older, had internet access and accepted the informed consent prior to the administration of the survey (Figure 1).

Table 1 shows that, out of the 1776 respondents, 57.9% (1028) were female, the median age was 24 (interquartile range: 20–28 years), the majority had university education (67.5%), had no chronic diseases (80.4%), were not healthcare workers (89.5%), had a median of 4 people at home (interquartile range: 3–5 people), had not yet been infected with COVID-19 (61.1%), nor had their family members (45.3%) and they were planning to be vaccinated (70.4%).

The collected data indicated significant differences in the acceptance of the COVID-19 vaccine in the different departments of Peru. Departments in the central and south highlands of the country had the highest percentages of people who did not plan to be vaccinated or were hesitant to be vaccinated (Figure 2).

Among the reasons indicated by respondents who did not plan to be vaccinated by a possible COVID-19 vaccine, the most frequent was the fear of side effects (60%), followed by believing that “they are experimenting on me” (34%) and to a lesser extent “I may get sick with COVID-19” (6%) (Figure 3).

As shown on Table 2, the bivariate analysis between the sociodemographic characteristics and the intent to get vaccinated, differences were found in regard to age (*p* = 0.021), educational level (*p* < 0.001), being a healthcare worker (*p* < 0.001), having been sick with COVID-19 (*p* < 0.001) or if their family or friends got sick (*p* = 0.003).

As shown on Table 3, when multivariate analysis (adjusted by department of residence) was performed, it was found that those who had a higher frequency of not wanting or not knowing if they were going to be vaccinated were those who did not know if they had been infected with COVID-19 (aPR: 1.40; 95% CI: 1.09–1.81; *p*-value = 0.008). In contrast, there was a lower frequency of refusal to the vaccine among university students (aPR: 0.75; 95% CI: 0.61–0.92; *p*-value = 0.005) and healthcare workers (aPR: 0.59; 95% CI: 0.44–0.80; *p*-value = 0.001).

## 4. Discussion

Evidence suggests that the next global challenge, after successfully developing a vaccine against COVID-19, will be to persuade a sufficient proportion of the inhabitants to accept vaccination, and thus mitigate the impact of the virus around the world [77,78]. It has been reported that the willingness to get vaccinated against COVID-19 in Kuwait was positively influenced by younger age, being male, having a higher education level, having been previously vaccinated against seasonal influenza, being a healthcare worker and working in the private sector [79]. Similar results were observed in a study that assessed the COVID-19 vaccine hesitancy among Ethiopian healthcare workers [80], among undergraduate students from central and southern Italy [81] and among the Chinese population [82]. In the latter study it was observed that the COVID-19 vaccine hesitancy was modest in China [82]. A study that evaluated the willingness to get vaccinated against COVID-19 in Burkina Faso, Ethiopia, Malawi, Mali, Nigeria and Uganda reported that four in five people were willing to get vaccinated, except in Ethiopia [83]. It was found that the main reason for this discrepancy in Ethiopia was because of the potential side effects of the vaccine [83]. This was also observed in a study performed in Saudi Arabia where the vaccine refusers indicated that their safety concerns would only be dissipated with more reported clinical studies [84]. These concerns have been related to belief in conspiracy theories [85] and the wait-and-see approach to assess possible long-term health risks [86].

According to previously published studies related to the intention, associated predictors and attitude towards vaccination against COVID-19 in Peru, it was found that adults and older adults are the most willing to accept the vaccine [87]. However, other factors were also found to be associated with vaccination against COVID-19. For example, in the study by Herrera et al., a higher prevalence of intention to vaccinate was found in people with COVID-19 symptoms, economic insecurity, fear that they or a family member would get sick and those who received recommendations from family members, friends, health workers, the World Health Organization (WHO) or government officials [88]. On the other hand, in a study carried out in Arequipa where 120 people who went to a popular market were surveyed, it was found that people over 25 years of age and with a higher level of education had higher intention to get vaccinated [89]. In the Caycho et al. study, the results indicated that the most important predictor for getting vaccinated against COVID-19 was confidence in vaccines [90]. In addition, being over 65 years old, residing in Lima, being afraid of getting sick, spreading or dying from COVID-19, perceiving a greater probability or severity of contracting the disease, thinking of or receiving information about COVID-19, suffering from anxiety and having postponed a previous vaccination was associated with a greater intention to get vaccinated [90]. Moreover, in the study by Serpa et al., adults and the elderly who were well informed about vaccination and had a higher level of education, especially educators, healthcare personnel and education students, were identified with a favorable attitude towards vaccination against COVID-19 [91].

Through this study in Peru, we found that seven out of ten respondents are willing to be vaccinated against COVID-19. That is, three out of ten respondents were unwilling or hesitant to be vaccinated. These results is similar to the current 65% of people vaccinated with both doses in Peru [50]. We also identified multiple independent variables associated with vaccine acceptance, such as a level of education, which was previously reported in the United States [75,78]. This could be a possible explanation for the high acceptance rate compared to the United Kingdom, Australia, the United States, Qatar and China, which are also within the herd immunity threshold, estimated to be between 55 and 82% to stop infection [75,77,78,92,93,94,95]. This finding is consistent with the results of previous studies in the United States, where it was found that having more years of education implied an increased acceptance of the COVID-19 vaccine [75,78,95,96]. That could be explained by the fact that these individuals have greater accessibility to information related to the vaccine, where the group with postgraduate education had a 75% willingness to vaccinate compared to those with a lower level of education [75]. Our results show that healthcare workers were the ones who most wanted to be vaccinated, as they were less likely to refuse the vaccine. It has been reported that healthcare workers see the possibility that being immunized would prevent the severe form of disease caused by COVID-19 [78,92,95,96,97]. Educating the population about the importance of COVID-19 vaccination, especially vulnerable populations, will require the effort of the government in conjunction with healthcare workers [75,98]. Therefore, hesitancy and resistance to vaccination should be addressed as early as possible by health science professionals [77,78,96,99].

Other determinants affecting vaccine acceptance were self-exposure to SARS-CoV-2 virus by those who did not know or doubted that they had been infected; thus, they were less interested in being vaccinated. In our study, we identified considerable differences in the desire to get vaccinated in several departments in the central highlands and some in the northern part of the country, who did not want to be vaccinated, which could be due to the inequality gap existing in the different regions of Peru. The differences between departments require differentiated responses from a territorial approach by the authorities [75,95,100]. Supporting this, a World Bank study indicated that inequality factors such as educational level, stable salary jobs, access to Internet and gender have contributed to the existing gap. This could be an explanation for the different opinions obtained in the study and the great impact of the pandemic in such places [75,78,95]. It is important to highlight that there is a small part of the population who are skeptical in regard of vaccination in general and not only of a new vaccine against COVID-19 [97]. This poses a risk to individuals and their community, since exposure to a contagious disease puts people at risk due to its rapid ability to spread if a significant portion of the population is not vaccinated [75,77,78,96,101]. For these reasons, a state approach to regions with lower vaccine acceptance rates would be appropriate, through different strategies, such as promoting access to basic health and education services, since a large part of the population is not in a position to comply with the provisions recommended by the country [77,78,95]. Even though 70% of the Peruvian population has received both COVID-19 vaccines, multiple regions of Peru reported lower vaccination rates, especially in the central highlands and northern part of the country [102], which matches with our results. Because of this sociodemographic discrepancies in COVID-19 vaccination in Peru, the government of announced that since 10 December 2021 it is mandatory for adults older than 18 years old to present a vaccination record with both COVID-19 vaccines doses to access all closed public spaces, excluding pharmacies and hospitals [103]. Furthermore, since the implementation of this measure a high number of non-vaccinated and incomplete vaccinated people filled vaccination centers across multiple regions in Peru, which could allow the country to reach the target 80% of its population to be vaccinated before the end of 2021 [104].

### Limitations

Regarding limitations, the fidelity and veracity of the data were considered since this is a study with a very important subjective component. We depend on the willingness of the population to collaborate. In addition, as this is a cross-sectional observational study, we do not have the capacity to make a prospective and retrospective analysis of the situation. On the other hand, we do not know if past experiences can influence people’s decision on the acceptance of the vaccine, since different phases of the pandemic lead to different responses to the situation [94,97]. This was the case in Italy, where it was found that as the perception of risk increased, more people were open to the possibility of being vaccinated against COVID-19 [75,78,94,97]. Additionally, our study focused on a sample of the population; this sample is representative of the general adult population and does not include members of the public who are institutionalized (e.g., incarcerated), or difficult to reach such as homeless people or those without internet access. The inability to reach these sectors limits the generalizability of our results. Finally, vulnerable groups such as children and the elderly were not taken into account [77]. We recommend considering these sectors for future studies. Another limitation is bias that occurred as a result of the cross-sectional study design to determine definitive cause and effect associations. Similarly, the responders performed a self-reported assessment in an online data collection platform, which could lead to under- or over-reporting, and the data collector does not have the ability to verify or validate.

It should be considered that the intention to receive the vaccine in this study does not necessarily translate into the actual number of people who will accept or decline a vaccination. For this reason, it is advisable to continue with health promotion in the different departments of Peru [78,104]. Despite these limitations, our findings provide important evidence regarding the level of acceptance and resistance to COVID-19 vaccination in a general population sample. Our study highlights the importance of understanding the various social, economic, political and psychological factors that contribute to hesitancy and resistance to COVID-19 vaccination, and how they can be used to maximize the positive public health effect. We would like to encourage more researchers to conduct similar studies using this population in order to better understand vaccine acceptance.

## 5. Conclusions

Acceptance of the COVID-19 vaccine in Peru is influenced by the level of education. This acceptance is relatively high in people with university education but decreases in people with high school or a lower level of education. If the educational level is an important factor, the government should provide greater accessibility to information and education about vaccines. In addition, given that healthcare workers are the sector that most want to be vaccinated, it is also important that they work together with governments to educate the population about the importance of vaccination.

## Figures and Tables

**Figure 1 vaccines-10-00048-f001:**
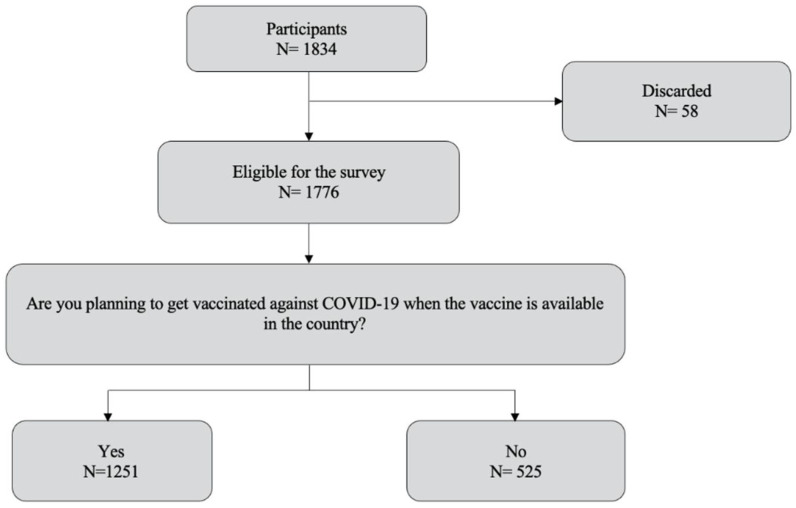
Flowchart of the survey respondents.

**Figure 2 vaccines-10-00048-f002:**
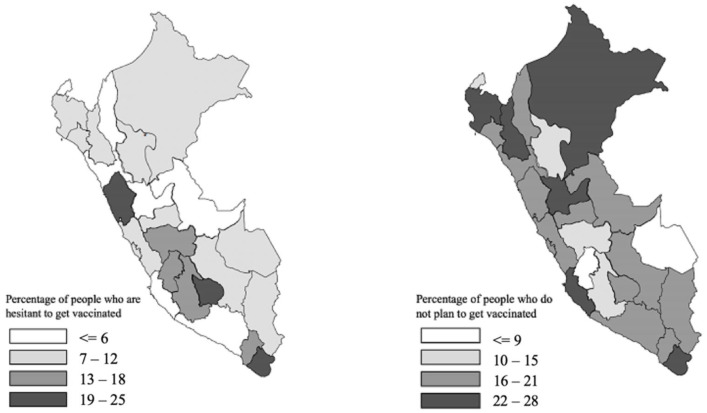
Geographical distribution of people that are hesitant or do not plant to get vaccinated in Peru.

**Figure 3 vaccines-10-00048-f003:**
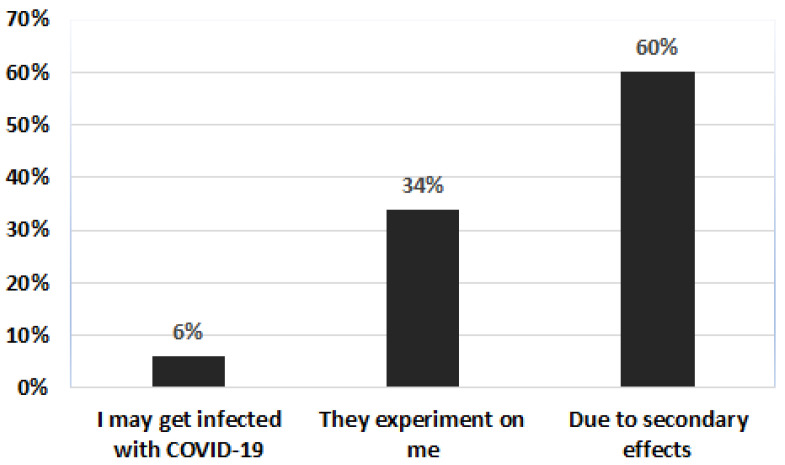
Reasons why people do not plan to be vaccinated against COVID-19 in Peru.

**Table 1 vaccines-10-00048-t001:** Sociodemographic characteristics of respondents.

Variable	Frequency	Percentage/Interquartile Range
**Gender**		
Male	748	42.1%
Female	1028	57.9%
**Age (years of age) ^a^**	24	20–28
**Level of education**		
High school or lower	145	8.2%
Technical	241	13.6%
University	1199	67.5%
Postgraduate	191	10.8%
**Household size ^a^**	4	3–5
**Healthcare worker**		
No	1589	89.5%
Yes	187	10.5%
**With a chronic disease**		
No	1428	80.4%
Yes	348	19.6%
**Got infected with COVID-19**?		
No	1085	61.1%
Yes (confirmed with a test)	244	13.7%
Yes (without a test)	112	6.3%
I don’t know/it is possible	335	18.9%
**Family or friends got infected with COVID-19**?		
No	805	45.3%
Yes (confirmed with a test)	657	37.0%
Yes (without a test)	156	8.8%
I don’t know/it is possible	158	8.9%
**Will you get vaccinated against COVID-19**?		
No	179	10.1%
Yes	1251	70.4%
I don’t know yet.	346	19.5%

^a^ Median and interquartile range are shown.

**Table 2 vaccines-10-00048-t002:** Bivariate analysis of sociodemographic characteristics and the intent to get vaccinated against COVID-19 in Peru.

Variable	Will You Get Vaccinated?			*p*-Value
	No	Yes	I Don’t Know	
**Gender**				
Male	72 (9.6%)	539 (72.1%)	137 (18.3%)	0.439
Female	107 (10.4%)	712 (69.3%)	209 (20.3%)	
**Age (years of age) ^a^**	29 (21–42)	24 (20–37)	23 (20–39)	0.021
**Level of education**				
High school or lower	16 (11.0%)	89 (61.4%)	40 (27.6%)	<0.001
Technical	20 (8.3%)	175 (72.6%)	46 (19.1%)	
University	107 (8.9%)	856 (71.4%)	236 (19.7%)	
Postgraduate	36 (18.9%)	131 (68.6%)	24 (12.5%)	
**Household size ^a^**	4 (3–5)	4 (3–5)	4 (3–5)	0.456
**Healthcare worker**				
No	167 (10.5%)	1096 (69.0%)	326 (20.5%)	<0.001
Yes	12 (6.4%)	155 (82.9%)	20 (10.7%)	
**With a chronic disease**				
No	152 (10.6%)	988 (69.2%)	288 (20.2%)	0.058
Yes	27 (7.8%)	263 (75.6%)	58 (16.6%)	
**Got infected with COVID-19**?				
No	113 (10.4%)	798 (73.6%)	174 (16.0%)	<0.001
Yes (confirmed with a test)	17 (7.0%)	169 (69.3%)	58 (23.7%)	
Yes (without a test)	12 (10.7%)	85 (75.9%)	15 (13.4%)	
I don’t know/it is possible	37 (11.0%)	199 (59.4%)	99 (29.6%)	
**Family or friends got infected with COVID-19**?				
No	81 (10.1%)	588 (73.0%)	136 (16.9%)	0.003
Yes (confirmed with a test)	66 (10.0%)	457 (69.6%)	134 (20.4%)	
Yes (without a test)	17 (10.9%)	113 (72.4%)	26 (16.7%)	
I don’t know/it is possible	15 (9.5%)	93 (58.9%)	50 (31.6%)	

^a^ Median and interquartile range are shown, *p*-values were calculated through the Kruskall–Wallis test. The other *p*-values were calculated with the Chi-squared test.

**Table 3 vaccines-10-00048-t003:** Bivariate and multivariate analysis of sociodemographic characteristics and the intent to get vaccinated against COVID-19 in Peru.

Variable	aPR (95% Confidence Internal) *p*-Value	
	Bivariate Analysis	Multivariate Analysis
**Female**	1.10 (0.95–1.27) 0.186	Did not use the model
**Age (years of age)**	1.00 (0.99–1.01) 0.459	Did not use the model
**Level of education**		
High school or lower	Ref.	Ref.
Technical	0.71 (0.49–1.01) 0.062	0.72 (0.49–1.04) 0.077
University	0.74 (0.60–0.92) 0.007	0.75 (0.61–0.92) 0.005
Postgraduate	0.81 (0.55–1.21) 0.312	0.93 (0.63–1.37) 0.724
**Household size**	1.00 (0.96–1.05) 0.965	Did not use the model
**Healthcare worker**	0.55 (0.41–0.74) <0.001	0.59 (0.44–0.80) 0.001
**With a chronic disease**	0.79 (0.58–1.09) 0.148	Did not use the model
**Got infected with COVID-19**?		
No	Ref.	Ref.
Yes (confirmed with a test)	1.16 (0.93–1.45) 0.179	1.15 (0.89–1.49) 0.289
Yes (without a test)	0.91 (0.62–1.34) 0.640	0.88 (0.56–1.39) 0.595
I don’t know/it is possible	1.53 (1.23–1.91) <0.001	1.40 (1.09–1.81) 0.008
**Family or friends got infected with COVID-19**?		
No	Ref.	Ref.
Yes (confirmed with a test)	1.13 (0.92–1.39) 0.256	1.04 (0.82–1.32) 0.744
Yes (without a test)	1.02 (0.72–1.46) 0.902	0.96 (0.65–1.41) 0.827
I don’t know/it is possible	1.53 (1.09–2.15) 0.015	1.22 (0.83–1.78) 0.310

## Data Availability

The data presented in this study are available on request from the corresponding author.

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
