# Peer review of "Sociodemographic Predictors Associated with the Willingness to Get Vaccinated against COVID-19 in Peru: A Cross-Sectional Survey"

_vaccines, 2021, doi:10.3390/vaccines10010048_

Round 1

Reviewer 1 Report

An analytical cross-sectional study using an anonymous survey in the 25 departments of Peru was performed. Educational level, whether being a healthcare worker and infected with COVID-19 were essential socio-demographic predictors on willingness to get vaccinated in Peru. The study is meaningful and essential on future COVID-19 vaccination policies. However, the following questions should be answered.

  1. The author should demonstrate whether pilot study on the questionnaire was performed or not?
  2. How did the author test the validity of the questionnaire on the willingness to get vaccinated against COVID-19?
  3. Based on socio-demographic characteristic with the willingness to get vaccinated against COVID-19, potential and relevant measures to improve the vaccination rate should be discussed.
  4. As a series of studies on the willingness to vaccinate against COVID-19 have been published, the differences and association of this study with other researches need be further discussed.

Author Response

We thank the reviewer for the positive review of our manuscript. The reviewer has made some critical and insightful comments that have definitely improved the final version. We have carefully amended the paper as suggested by the reviewer.

Comments

An analytical cross-sectional study using an anonymous survey in the 25 departments of Peru was performed. Educational level, whether being a healthcare worker and infected with COVID-19 were essential socio-demographic predictors on willingness to get vaccinated in Peru. The study is meaningful and essential on future COVID-19 vaccination policies. However, the following questions should be answered.

  1. The author should demonstrate whether pilot study on the questionnaire was performed or not?

We thank the reviewer for this comment, we have added the following paragraph to include the pilot study: “The survey was initially evaluated by 10 expert judges using Aiken's V. After including the experts’ observations, a pilot study was performed on the second week of November 2020 in the 25 departments of Peru. The pilot data was used to calculate the minimal sample size necessary for the actual study.”

  1. How did the author test the validity of the questionnaire on the willingness to get vaccinated against COVID-19?

We thank the reviewer for this comment. As we described it in the Variable sub-section of the Materials and Methods section we have utilized a previously published instrument (https://doi.org/10.1016/j.eclinm.2020.100495), which was adapted and translated to Spanish. As mentioned in the original paper, the instrument was internally validated by using

bootstrap resampling (1,000 samples) and the area under the curve value was calculated for model optimism. Since this survey was performed for the United States population, and as described in the paper, we performed a cultural validation through a report that evaluated each question from the survey on three criteria: relevance, coherence and clarity. The report was filled by a specialist in Social Sciences and then the survey was sent to the population.

  1. Based on socio-demographic characteristic with the willingness to get vaccinated against COVID-19, potential and relevant measures to improve the vaccination rate should be discussed.

The reviewer brings and excellent point and we have included the recent measures imposed in Peru to improve the vaccination rate. We have added the following statements in the Discussion: “Even though, 70% of the Peruvian population has received both COVID-19 vaccines, multiple regions of Peru reported lower vaccination rates, especially in the central highlands and northern part of the country, which matches with our results. Because of this socio-demographic discrepancies in COVID-19 vaccination in Peru, the government of announced that since December 10, 2021 it is mandatory for adults older than 18 years old to present the vaccination record with both COVID-19 vaccines doses to access all closed public spaces, excluding pharmacies and hospitals. Furthermore, since the implementation of this measure a high number of non-vaccinated and incomplete vaccinated people filled vaccination centers across multiple regions in Peru, which could allow the country to reach the target 80% of its population to be vaccinated before the end of 2021.”

  1. As a series of studies on the willingness to vaccinate against COVID-19 have been published, the differences and association of this study with other researches need be further discussed.

The reviewer brings an excellent point and before we discuss the previous reported studies in Peru, we have added discussion about willingness to vaccinate in different countries. Thus, we have added the following statements in the Discussion: “It has been reported that the willingness to get vaccinated against COVID-19 in Kuwait was positively influenced by younger age, being male, having a higher education level, vaccinated against seasonal influenza previously, being a healthcare worker, and working in the private sector. Similar results were observed in a study that assessed the COVID-19 vaccine hesitancy among Ethiopian healthcare workers, among undergraduate students from central and southern Italy,  and among the Chinese population. In the latter study it was observed that the COVID-19 vaccine hesitancy was modest in China. A study that evaluated the willingness to get vaccinated against COVID-19 in Burkina Faso, Ethiopia, Malawi, Mali, Nigeria and Uganda reported that four in five people were willing to get vaccinated, except in Ethiopia. It was found that the main reason for this discrepancy in Ethiopia was because the potential side effects of the vaccine. This was also observed in a study performed in Saudi Arabia where the vaccine refusers indicated that their safety concerns would only get dissipated with more reported clinical studies. These concerns have been related to belief in conspiracy theories and the wait-and-see approach to assess possible long-term health risks.”

Reviewer 2 Report

First of all, I would like to thank for the opportunity to review this paper. COVID-19 is an ongoing pandemic that has resulted in global health, economic and social crises. Actually, the vaccination campaign is the first method to counteract the COVID-19 pandemic; however, sufficient vaccination coverage is conditioned by the people’s acceptance of these vaccines in the general population. In this context, the paper under review is aimed at determine the socio-demographic predictors associated with the willingness of getting vaccinated against COVID-19 in Peru.

The article is interesting and may provide important information for public health, but it must be improved.

Title: it is overstated, I suggest to better identify the target of the study that is a sample of people.

Introduction: It is redundant, it should be shortened. The authors should make it clear about what is the gap in the literature that is filled with this study, considering the actual epidemiology. What is the international situation regarding the acceptance of the vaccination in the adult population (refer to articles with DOI: https://doi.org/10.3390/vaccines9060638). What is the possible contributions of the study to the literature? What are the possible practical implications of the study?

Methods: The enrolment procedure must be better specified, is seems a little confusing who was involved in the survey? How did the authors choose the way used to enroll their sample? How did they avoid the selection bias, since a snowball sampling was used? What is the reference population?

About the questionnaire, it is mentioned a validation but methodology and results are not reported. What about face validity, reliability and intelligibility?

Statistical analysis: I suggest to insert a measure of the magnitude of the effect for the comparisons. Please consider to include effect sizes.

Discussion: I suggest to emphasize the contribution of the study to the literature. The implications and recommendations based on previous experience in other population groups should be reported, because it seems a study with a unique local impact. The authors conclude that “the government should provide greater accessibility to information and education about vaccines” but the effectiveness of the information strategy should be taken into account (refer to articles with DOI: https://doi.org/10.3390/vaccines9060638).

Author Response

We thank the reviewer for the positive review of our manuscript. The reviewer has made some critical and insightful comments that have definitely improved the final version. We have carefully amended the paper as suggested by the reviewer.

Comments

First of all, I would like to thank for the opportunity to review this paper. COVID-19 is an ongoing pandemic that has resulted in global health, economic and social crises. Actually, the vaccination campaign is the first method to counteract the COVID-19 pandemic; however, sufficient vaccination coverage is conditioned by the people’s acceptance of these vaccines in the general population. In this context, the paper under review is aimed at determine the socio-demographic predictors associated with the willingness of getting vaccinated against COVID-19 in Peru. The article is interesting and may provide important information for public health, but it must be improved.

  1. Title: it is overstated, I suggest to better identify the target of the study that is a sample of people.

We thank the reviewer for this comment; however, we consider that the title already includes the wording: A cross-sectional survey that details the type of study that was performed. Furthermore, it is clearly stated in the abstract that this is sample of 1776 respondents in the 25 departments of Peru and there is Limitations sub-section that details that this study only represents a sample of the population. This can be seen in the following statements: “Also, our study focused on a sample of the population; this sample is representative of the general adult population and does not include members of the public who are institutionalized (e.g., incarcerated), or difficult to reach such as homeless people or without internet access. The inability to reach these sectors limits the generalizability of our results. Finally, vulnerable groups such as children and the elderly were not taken into account. We recommend considering these sectors for future studies.” Therefore, we have decided to maintain the title as it is.

  1. Introduction: It is redundant, it should be shortened. The authors should make it clear about what is the gap in the literature that is filled with this study, considering the actual epidemiology.

We thank the reviewer for this comment; however, we consider that the introduction is appropriate since we decided to structure it in a way that the reader can understand first the most relevant published articles related to COVID-19 in Peru, then present the existing disinformation present in the country. We also summarized the implement regulations related to the vaccination plan in Peru, the vaccine purchases and the Vacunagate scandal that directly impacted the willingness to vaccinate. Last, we have included current statistics of the vaccination rates in Peru and present the gap in literature in the last part of the Introduction. We have included the following statements in the Introduction: “The importance of this issue is reflected in the vaccination process itself, especially after knowing that some sectors of the population refuse to get vaccinated in countries where the process has already begun. In Peru, vaccination is already undergoing for people over 12 years old and a third vaccine dose is been administered to adults over 18 years who received the second dose at least 5 months before. As of November 26, 2021, 17.8 million people in Peru has received both doses of the COVID-19 vaccine, constituting about 65% of the intended 27.4 million people to be vaccinated. Therefore, it is necessary to determine the factors associated for people in Peru not wanting to be vaccinated. The general objective of our study was to determine the socio-demographic predictors associated with the willingness of getting vaccinated against COVID-19 in Peru.” In the discussion we have also included information related to the recently imposed mandate for adults older than 18 years old to present the vaccination record with both COVID-19 vaccines doses to access all closed public spaces, excluding pharmacies and hospitals.

  1. What is the international situation regarding the acceptance of the vaccination in the adult population (refer to articles with DOI: https://doi.org/10.3390/vaccines9060638).

The reviewer brings an excellent point and before we discuss the previous reported studies in Peru, we have added discussion about willingness to vaccinate in different countries including the recommended reference. Thus, we have added the following statements in the Discussion: “It has been reported that the willingness to get vaccinated against COVID-19 in Kuwait was positively influenced by younger age, being male, having a higher education level, vaccinated against seasonal influenza previously, being a healthcare worker, and working in the private sector. Similar results were observed in a study that assessed the COVID-19 vaccine hesitancy among Ethiopian healthcare workers, among undergraduate students from central and southern Italy,  and among the Chinese population. In the latter study it was observed that the COVID-19 vaccine hesitancy was modest in China. A study that evaluated the willingness to get vaccinated against COVID-19 in Burkina Faso, Ethiopia, Malawi, Mali, Nigeria and Uganda reported that four in five people were willing to get vaccinated, except in Ethiopia. It was found that the main reason for this discrepancy in Ethiopia was because the potential side effects of the vaccine. This was also observed in a study performed in Saudi Arabia where the vaccine refusers indicated that their safety concerns would only get dissipated with more reported clinical studies. These concerns have been related to belief in conspiracy theories and the wait-and-see approach to assess possible long-term health risks.”

  1. What is the possible contributions of the study to the literature? What are the possible practical implications of the study?

We thank the reviewer for this comment, the last paragraph of the Limitations sub-section include the contribution and practical implications of the study: “It should be considered that the intention to receive the vaccine in this study does not necessarily translate into the actual number of people who will accept or decline to be vaccinated. For this reason, it is advisable to continue with health promotion in the different departments of Peru. Despite these limitations, our findings provide important evidence regarding the level of acceptance and resistance to COVID-19 vaccination in a general population sample. Our study highlights the importance of understanding the various social, economic, political, and psychological factors that contribute to hesitancy and resistance to COVID-19 vaccination, and how they can be used to maximize the positive public health effect. We would like to encourage more researchers to conduct similar studies using this population in order to better understand vaccine acceptance.”

  1. Methods: The enrolment procedure must be better specified, is seems a little confusing who was involved in the survey? How did the authors choose the way used to enroll their sample? How did they avoid the selection bias, since a snowball sampling was used? What is the reference population?

We thank the reviewer for these observations. We have added additional statements in the Procedures sub-section in the Materials and Methods section to clarify about the enrollment. We have added that it was oriented to the population over 18 years old and that we contacted healthcare personnel to ask for collaboration and dissemination of the survey, providing informed consent where the confidentiality of the information is assured. We have added an additional statement related to bias: “Another limitation is bias occurred as a result of the cross-sectional study design to determine definitive cause and effect associations. Similarly, the responders performed a self-reported assessment in an online data collection platform, which could lead to under or over-reporting and the data collector has not ability to verify or validate.”

  1. About the questionnaire, it is mentioned a validation but methodology and results are not reported. What about face validity, reliability and intelligibility?

We thank the reviewer for this comment. As we described it in the Variable sub-section of the Materials and Methods section we have utilized a previously published instrument (https://doi.org/10.1016/j.eclinm.2020.100495), which was adapted and translated to Spanish. As mentioned in the original paper, the instrument was internally validated by using

bootstrap resampling (1,000 samples) and the area under the curve value was calculated for model optimism. Since this survey was performed for the United States population, and as described in the paper, we performed a cultural validation through a report that evaluated each question from the survey on three criteria: relevance, coherence and clarity. The report was filled by a specialist in Social Sciences and then the survey was sent to the population.

  1. Statistical analysis: I suggest to insert a measure of the magnitude of the effect for the comparisons. Please consider to include effect sizes.

We thank the reviewer for the suggestion, but cross-sectional studies cannot establish a cause-and-effect relationship or analyze behavior over a period of time. To investigate cause and effect, you need to do a longitudinal study or an experimental study. We have added a statement in the Limitations sub-section related to bias.

  1. Discussion: I suggest to emphasize the contribution of the study to the literature. The implications and recommendations based on previous experience in other population groups should be reported, because it seems a study with a unique local impact. The authors conclude that “the government should provide greater accessibility to information and education about vaccines” but the effectiveness of the information strategy should be taken into account (refer to articles with DOI: https://doi.org/10.3390/vaccines9060638).

The reviewer brings an excellent point and before we discuss the previous reported studies in Peru, we have added discussion about willingness to vaccinate in different countries, including the recommended reference. Thus, we have added the following statements in the Discussion: “It has been reported that the willingness to get vaccinated against COVID-19 in Kuwait was positively influenced by younger age, being male, having a higher education level, vaccinated against seasonal influenza previously, being a healthcare worker, and working in the private sector. Similar results were observed in a study that assessed the COVID-19 vaccine hesitancy among Ethiopian healthcare workers, among undergraduate students from central and southern Italy,  and among the Chinese population. In the latter study it was observed that the COVID-19 vaccine hesitancy was modest in China. A study that evaluated the willingness to get vaccinated against COVID-19 in Burkina Faso, Ethiopia, Malawi, Mali, Nigeria and Uganda reported that four in five people were willing to get vaccinated, except in Ethiopia. It was found that the main reason for this discrepancy in Ethiopia was because the potential side effects of the vaccine. This was also observed in a study performed in Saudi Arabia where the vaccine refusers indicated that their safety concerns would only get dissipated with more reported clinical studies. These concerns have been related to belief in conspiracy theories and the wait-and-see approach to assess possible long-term health risks.”

Furthermore, we  have expanded on our statement “the government should provide greater accessibility to information and education about vaccines” to include actual recent measures that the government has implemented to increase vaccination rates. The following statements have been included in the Discussion section: “For these reasons, a state approach to regions with lower vaccine acceptance rates would be appropriate, through different strategies, such as promoting access to basic health and education services, since a large part of the population is not in a position to comply with the provisions recommended by the country. Even though, 70% of the Peruvian population has received both COVID-19 vaccines, multiple regions of Peru reported lower vaccination rates, especially in the central highlands and northern part of the country, which matches with our results. Because of this socio-demographic discrepancies in COVID-19 vaccination in Peru, the government of announced that since December 10, 2021 it is mandatory for adults older than 18 years old to present the vaccination record with both COVID-19 vaccines doses to access all closed public spaces, excluding pharmacies and hospitals. Furthermore, since the implementation of this measure a high number of non-vaccinated and incomplete vaccinated people filled vaccination centers across multiple regions in Peru, which could allow the country to reach the target 80% of its population to be vaccinated before the end of 2021.”

Round 2

Reviewer 2 Report

The paper was improved according my comments and it is now suitable for publication